# Gonadectomy in Raccoons: Anesthetic and Cardiorespiratory Effects of Two Ketamine-Based Pre-Anesthetic Protocols before Sevoflurane-Sufentanil

**DOI:** 10.3390/ani10112110

**Published:** 2020-11-13

**Authors:** Sara Nannarone, Valentina De Monte, Rolando Arcelli, Laura Menchetti, Rodolfo Gialletti

**Affiliations:** 1Department of Veterinary Medicine, Perugia University, 06126 Perugia, Italy; valedemo78@gmail.com (V.D.M.); laura.menchetti7@gmail.com (L.M.); rodolfo.gialletti@unipg.it (R.G.); 2Department of Agricultural and Food Sciences, University of Bologna, Viale Fanin 44, 40127 Bologna, Italy

**Keywords:** anesthesia, blood gas analysis, dexmedetomidine, ketamine, raccoon, sevoflurane, sufentanil

## Abstract

**Simple Summary:**

Raccoon (*Procyon lotor*) is a non-indigenous wildlife species originating from America which was introduced to Europe in the twentieth century. Raccoons are potential vectors of many diseases that can threaten human and domestic animals’ health. The Northern raccoon is listed within the worst 100 invasive alien species in Europe and since 2004 it has also been introduced to northern Italy. Containment, including birth control and eradication have become crucial steps to preserve and maintain autochthone biodiversity. The aim of this study was to find two anesthetic drug combinations suitable for a safe chemical immobilization and to compare them for qualitative anesthetic effects and influence on vital clinical parameters. The procedure was required to allow a safe handling of wild raccoons, confiscated from northern Italy and referred to the veterinary teaching hospital for gonadectomy, microchip application and subsequent re-release into a protected area.

**Abstract:**

Nineteen raccoons were enrolled in this study. The aim was to evaluate and compare the quality of anesthesia and the cardiorespiratory effects following treatment with a ketamine-based combination with either dexmedetomidine (KD group) or midazolam (KM group) in raccoons undergoing ovariohysterectomy/orchiectomy. General anesthesia was maintained with sevoflurane in oxygen and sufentanil infusion. The time required to approach the animals was similar among groups resulting in a median of 5 min after IM injection. Animals in group KD were scored with greater myorelaxation (*p* < 0.01) and easier intubation (*p* < 0.05). Moreover 70% of them did not require other drugs for tracheal intubation unlike animals in group KM, which required propofol in 100% of cases. After intubation and connection to the breathing circuit, physiological parameters were monitored continuously and recorded every 5 min. Sevoflurane requirements were lower in group KD than KM (*p* = 0.005). Blood pressure was maintained within physiological ranges in both groups but with higher values in group KM (*p* < 0.05). Mild respiratory depression occurred during surgery and animals in group KM showed greater respiratory acidosis (*p* < 0.05). Recovery was smooth and uneventful in all animals. Both anesthetic protocols can be recommended for safe anesthesia in wild raccoons.

## 1. Introduction

Raccoons (*Procyon lotor*) are originally from North and Central America, and are also kept as zoo animals and their popularity as pets has increased in several countries worldwide.

In Italy, the raccoon is included in the list of animal species that can pose a danger to public health and safety and its sale and private detention is prohibited, according to the Decree of the Ministry of the Environment of 19 April 1996 [1]. Raccoons are able to reproduce and spread easily, especially in their natural environment near rivers, canals or lakes in the area in which they were freed (i.e., in Lombardy region). Ecological, economic and health impacts can be easily seen. In particular, several autochthons animal species can suffer from excessive predation by raccoons, such as birds (predation of the nests) and some species of amphibians, with a substantial decrease in population size [2]. The raccoon can also affect agricultural and livestock activities and urban infrastructure. Finally, they are potential vectors of many pathogens that can threaten human and domestic animals’ health, such as the nematode *Baylisascaris procyonis*, whose larva migrans can lead to deadly encephalitis in humans [3]. In addition, raccoons are known to potentially carry rabies, distemper, histoplasmosis, trypanosomiasis and mange [4]. The appropriate management of these wild animals, possible vectors of zoonoses, is an important factor that is part of the multidisciplinary One Health approach, which is essential for properly controlling and managing zoonoses. In Italy, the Provincial Police and the former State Forestry Corps have captured raccoons in the territories of their respective jurisdiction; the animals were then brought to authorized centers responsible for legal detention and medical procedures.

Nowadays, the management of raccoons in overpopulated areas is a long-term, costly management strategy and requires stable funding; however, it is usually not very effective mostly due to insufficient knowledge of the entity of the animal population.

Chemical immobilization is an important tool for the safe and humane handling of wildlife, with several injectable and inhalation anesthetic options available for this scope [5,6,7]. Several drugs and drug combinations have been reported to provide successful immobilization in raccoons, including saffan [8], succinylcholine chloride [9], tiletamine-zolazepam [10,11] and ketamine [12,13,14]. Drug mixtures may have synergistic effects on induction of general anesthesia with better recovery times when compared to single drug use, and these effects may vary among species [15,16]. So far, there’s no standardized anesthetic protocol, described as safe for both the operators and the animals, in raccoon immobilization. Veterinarians usually rely on their personal experience or on anesthetic protocols extrapolated from domestic carnivores, however, from an animal welfare perspective, distress or unwanted side effects affecting vital clinical parameters or impairing recovery should be avoided when choosing anesthetic drugs.

The aim of this study was to evaluate the quality of anesthesia and the physiological parameters following a combination of either ketamine and dexmedetomidine or ketamine and midazolam in raccoons undergoing ovariohysterectomy/orchiectomy. The assessment included the quality of immobilization, the induction of general anesthesia and recovery time, as well as the cardiorespiratory effects of these drugs during maintenance of anesthesia with sevoflurane and sufentanil constant rate infusion (CRI).

## 2. Materials and Methods

### 2.1. Animals and Procedures

Nineteen raccoons, 10 females and 9 males, weighing between 4.2 and 10 kg (mean ± SD = 7 ± 1.6 kg) were enrolled in this study between mid-March and the beginning of June in 2014. They underwent ovariohysterectomy/orchiectomy at the Veterinary Teaching Hospital (VTH) of the University of Perugia according to an agreement within CITES (the Convention on International Trade in Endangered Species of Wild Fauna and Flora) between the Ministry for the Environment, Land Protection and State Forestry Corps and our Institution (prot. 991 03/2014), which included an ethical statement, for a management program of established populations to prevent expansion of the species. Thus, statistical power analysis was not applied, and the sample size was based on the requirements of the State Forestry Corps project and the issued permits [17].

Twenty-four hours before the procedure, the animals were hospitalized at the VTH. Food and water were withheld for 12 h before surgery. On the day of surgery, the raccoons were weighed with a digital platform scale. Animals were identified as either juvenile or adult based on presence of external genitalia, body mass and teeth consumption [18]. The RANDBETWEEN function of Excel was used for allocation of animals into two groups to receive one of the following drugs combinations intramuscularly (IM): 7 mg/kg ketamine (Ketavet 100^®^, MSD Animal Health S.r.l., Segrate, MI, Italy) with either 7 µg/kg dexmedetomidine (Dexdomitor, 0.5 mg/mL, Elanco Animal Health, Eli Lilly Italia S.p.A., Sesto Fiorentino, FI, Italy) (KD group), or 0.3 mg/kg midazolam (Midazolam I.B.I., 0.5%^®^, Aprilia, LT, Italy) (KM group). All drugs were mixed in the same syringe and injected in the triceps muscle with the animal in a containment cage equipped with a squeeze chute (Figure 1).

The anesthetist responsible of the following assessment was blinded to the drugs mixture. Times (minutes) from injection to the first signs of sedation (‘inj-sed’) such as ataxia, weakness and head dropping, and to recumbency (‘inj-rec’) were recorded for each animal. If immobilization was not reached within 8 min, half of the initial dose was additionally administered IM. Time from injection to first manual approach was recorded (‘inj-approach’). Thereafter the animal was removed from the cage and the overall quality and the depth of anesthesia were assessed with a descriptive score scale (DSS) modified by Nannarone et al. 2020 [19]. The DSS included depth of anesthesia, muscle relaxation and ease of cephalic catheterization (Table 1). Palpebral reflex, response to tactile stimulus and tongue relaxation were also checked and recorded as present or not present. A cephalic vein was aseptically catheterized (Delta Ven^®^ 22 G, Delta Med S.p.A., Viadana, MN, Italy) and fluids administration was started (5 mL/kg/h 0.9% NaCl; S.A.L.F. S.p.A. Laboratorio Farmacologico, Cenate Sotto, BG, Italy).

Baseline vital parameters were recorded as soon as the animal was approachable outside of the cage and thereafter every 5 min from the beginning of general anesthesia until the end of the procedure.

If the level of anesthesia was adequate to allow endotracheal intubation, 0.2 mL of 2% lidocaine (Lidocaina 2%, Esteve SpA, Milan, MI, Italy) was delivered to the larynx with a mucosal atomization device (MADgic™, Wolfe Tory Medical, Inc., Salt Lake City, UT, USA) and an orotracheal tube was placed. Ease of intubation was scored according to the DSS (Table 1); if swallowing reflex and/or stiffness of the mandible persisted, intravenous (IV) propofol (Proposure, Merial SpA, Milan, MI, Italy) was administered to effect and the total amount required was recorded. After intubation, animals were connected to a non-rebreathing circuit and general anesthesia was maintained with sevoflurane (Sevoflo; Veterinaria Esteve, Esteve S.p.A., Milan, MI, Italy) in 100% oxygen (250 mL/kg/min) and a CRI of sufentanil (0.5 µg/kg/h; Disufen, A.C.R.A.F. S.p.A., Rome, Italy). Sevoflurane delivery was modified based on the depth of anesthesia by evaluating the responses to surgical stimulation. The surgical area was clipped and aseptically prepared, then the animals were transferred to the operating room and positioned in dorsal recumbency. The same team performed all the surgeries.

A 24 or 26 G catheter (Delta Ven^®^, Delta Med S.p.A., Viadana, MN, Italy) was aseptically placed percutaneously and secured in the femoral artery (Figure 2) to measure arterial blood pressure and obtain arterial blood samples for blood-gas analysis (i-STAT Portable Clinical Analyzer, Abbott, Burke & Burke, Assago, MI, Italy). Electrocardiogram, heart rate (HR, beats/min), respiratory rate (RR, breaths/min), arterial oxygen saturation of hemoglobin (SpO_2_, %), end-tidal carbon dioxide partial pressure (etCO_2_, mmHg), sevoflurane exhaled concentration (etSevo, %), invasive systolic, diastolic and mean arterial pressures (SAP, DAP, MAP, mmHg), and body temperature (T, °C), were monitored continuously and recorded every 5 min (HB100 multiparametric monitor, Foschi S.r.l., Rome, Italy) (Figure 2).

All the raccoons received IV cefazoline (Cefazolina TEVA, Teva Italia Srl, Milan, MI, Italy) 30 mg/kg before surgery and meloxicam (Metacam, Boeringher Ingelheim Italia S.p.A., Milan, MI, Italy) 0.2 mg/kg at the end of surgery.

At the end of anesthesia, the arterial line was removed with appropriate compression, 0.2 mg/kg atipamezole (Antisedan, Vétoquinol Italia S.r.l., Bertinoro, FC, Italy) (KD group) or an equal volume of normal saline (KM group) was administered IM by the anesthetist who was blinded to the drug administered. Times (min) between the end of anesthesia and re-appearance of palpebral reflex (‘end-palp’), extubation (‘end-ext’), lifting of the head (‘end-head’) and standing (‘end-stand’) were recorded as “recovery times”. Post-operative monitoring was carried out on all raccoons for the following 24 h and the release in the protected area was planned thereafter.

### 2.2. Statistical Analysis

Associations between groups and categorical variables were evaluated by Fisher’s exact and z-tests. Results were presented as numbers and percentages. Differences between groups (two levels: KD group and KM group) in body weight, variables measured as scores, ‘inj-sed’, ‘inj-rec’, ‘inj-approach’, “recovery times” and blood-gas values were compared using Mann–Whitney U Tests and expressed as medians (Mdn) and interquartile ranges (IQR). Mann–Whitney U Tests were also used to evaluate the effect of gender and age on these parameters. For repeated measurements (physiological parameters recorded every 5 min), data were analyzed by Linear Mixed Models in which animals and time points were included as subjects and repeated factors, respectively. The models evaluated the effects of time (seven levels: from 1 to 30 min after surgical incision, 5 min interval time), groups (two levels: KD group and KM group) and their interaction. Baseline values were included in the models as covariates. These results were expressed as estimated marginal means ± standard error (SE) [20]. Statistical analyses were performed with SPSS Statistics version 23 (IBM, SPSS Inc., Chicago, IL, USA) and GraphPad Prism, version 7.0 (GraphPad Software, San Diego, CA, USA). Statistical significance occurred when *p* ≤ 0.05 but trends (*p* < 0.1) were also presented and discussed.

## 3. Results

The study was completed without any major complications.

Two animals in group KM were excluded from statistical analysis: one due to the inability to place the arterial line and the other due to inappropriate IM needle placement during premedication, which led to only partial administration of the drugs. Table 2 shows the demographic data of subjects included in the study. No statistical difference was found between the groups in these parameters.

Two animals out of 10 (20.0%) and three animals out of 7 (42.9%) in group KD and KM respectively required an additional dose of anesthetic mixture for complete myorelaxation. The parameters related to the immobilization phase showed no statistical differences between groups KD and KM (*p* = 0.193, *p* = 0.364 and *p* = 0.161 for ‘inj-sed’, ‘inj-rec’ and ‘inj-approach’, respectively) and a high variability was found in both groups (Figure 3a). Regardless of the group, in half of the animals, the first signs of sedation were observed within 2 min from injection (IQR = 2–2 min) and recumbency was achieved within 4 min (IQR = 3–5 min). The time required to approach the animal after injection was 5 min (IQR = 4–10 min).

Conversely, differences were found in the recovery times (Figure 3b). In particular, the times between the end of anesthesia and appearance of palpebral reflex (‘end-palp’; *p* = 0.043) as well as the times between the end of anesthesia and extubation (‘end-ext’; *p* = 0.010) were higher in KD than KM group. Only a trend toward significance was found for ‘end-head’ (*p* = 0.088) and no differences for ‘end-stand’ (*p* = 0.138). Overall, median recovery times were 5 min for ‘end-palp’ (IQR = 4–7 min), 13 min for ‘end-ext’ (IQR = 8–15 min), 25 min for ‘end-head’ (IQR = 17–35 min), and 49 min for ‘end-stand’ (IQR = 37–68 min).

The recovery times were not influenced by gender and age.

Anaesthesia time was longer in group KM (Mdn = 65 min; IQR = 60–115 min) than in group KD (Mdn = 38 min; IQR = 35–55 min; *p* = 0.005).

Depth of anesthesia was scored as “profound” in eight (80.0%) and four (57.1%) animals, as “mild” in two (20.0%) and three (42.9%) animals in groups KD and KM, respectively (*p* > 0.1; Table 3). There was a significant difference in myorelaxation (*p* < 0.01) which was scored as “excellent” in 70% of the animals in KD group and in no animals (0.0%) in KM group. Myorelaxation was rated as “moderate” or “mild” in most animals of group KM (85.8%, Table 3). Furthermore, a significant difference was noted in ease of intubation, which was achieved without additional drugs in 70% of the raccoons in group KD while in group KM 100% of the animals required the use of propofol (*p* < 0.05; Table 3). The amount of propofol required for induction was 1.1 ± 0.2 and 1.4 ± 0.6 mg/kg in group KD and KM, respectively.

Sevoflurane requirements (etSev) differed among groups with an estimated marginal mean (±SE) of 1.4 ± 0.1 and 1.6 ± 0.1% in groups KD and KM, respectively (*p* = 0.005).

Statistically significant differences between groups were found in the marginal means of SAP (*p* < 0.01), MAP (*p* < 0.05) and DAP (*p* < 0.01), that were higher in group KM when compared to group KD. The temperature decreased over time from 36.0 ± 0.2 °C at the first time point to 35.1 ± 0.2 °C after 30 min (*p* < 0.05) but there were no statistically significant differences between groups. Influence of baseline values were only found for RR (*p* < 0.001), HR (*p* < 0.001) and etCO_2_ (*p* < 0.01; Table 4).

The mean values of pH, PaO_2_, PaCO_2_, SatO_2_, Hb, HCO_3_^-^, BE, TCO_2_, Hct, Na^+^, K^+^, and Ca^2+^ are reported in Table 5, and showed statistically significant differences for pH (*p* < 0.05) and PaCO_2_ (*p* < 0.05) values, showing a higher tendency to respiratory acidosis in group KM. No parameters were influenced by age or gender.

## 4. Discussion

The diffusion of invasive alien species to regions outside their native environment represents one of several threats to biodiversity conservation globally; veterinarians are more often than before called to manage these animals by implementing environmental protection strategies. It is therefore important to develop a safe and specific anesthetic protocol that will ensure the safety of the handler and the well-being of the wild animal suffering from the stress derived from being captured. Safe immobilization of raccoons in overpopulated areas can be required for a long-term management strategy. To the best of the authors knowledge this is the first study comparing anesthesia with ketamine–dexmedetomidine and ketamine–midazolam in raccoons. It is also the first study which evaluates intraoperative cardiorespiratory parameters in raccoons during sevoflurane–sufentanil CRI maintenance regimen, such as blood pressure and blood-gas analysis.

Ketamine hydrochloride is a non-barbiturate, phencyclidine derivate dissociative anesthetic agent with a large margin of safety. It has been used for chemical restraint in raccoons and likely is the most useful restraint agent in wildlife to date [21,22,23]. The wide margin of safety, the rapid onset of anesthetic effects and the fast recovery from anesthesia have made ketamine a common drug for use in field studies involving wild animals; nevertheless, optimum dosages of ketamine are very variable among and within species [14]. Ketamine can reduce aggressive behavior enabling safe handling while maintaining voluntary movement at low dosages [24]. It is a dissociative agent that causes sensory isolation in the brain while involuntary reflexes are maintained [25]. Ketamine used alone can cause somatic analgesia, hyperthermia, excessive salivation, catecholamine release, convulsions and poor muscle relaxation; therefore, it is often used in combination with myorelaxants and/or sedatives leading to reduced side effects and overall drug volume [26].

Dexmedetomidine is the newest α2-adrenoreceptor agonist representing the active enantiomer of the racemic mixture medetomidine, which lacks the pharmacologically inactive enantiomer levo-medetomidine. Dexmedetomidine is twice as potent in anesthetic efficacy than medetomidine and 40 times more effective than xylazine [27].

Midazolam is a benzodiazepine often used in combination with ketamine for its muscle relaxing properties, its cardiovascular sparing effects and predictable intramuscular absorption [28].

In veterinary medicine, the advantages of combining dexmedetomidine and ketamine have been related to the expected, rapid and smooth induction and maintenance of anesthesia. Dexmedetomidine provides excellent muscular relaxation and analgesia for surgical procedures, and the use of atipamezole offers rapid reversal of anesthetic effects leading to a fast recovery [19,27,29].

Our protocols included ketamine combined with either dexmedetomidine or midazolam, and atipamezole was administered at the end of the procedure to speed up recovery only when dexmedetomidine had been used, while the benzodiazepine was not antagonized.

The selected dose of ketamine was lower than that used in other studies, such as when combined with acepromazine at doses between 8–10 mg/kg [12] or with xylazine at doses from 10 [30] to 20 mg/kg [16]. However, our ketamine dose was higher than the 5.5 mg/kg used in association with 55 µg/kg medetomidine [23], and the 5 mg/kg combined with 50 µg/kg dexmedetomidine before alfaxalone and isoflurane [31].

Animals in the present study could be safely approached within 5 min from IM injection, which is earlier than the 15 min reported by Vogler et al. [31], and similar to the overall induction time of 6.4 ± 3.3 min reported by Robert et al. [23]. The faster onset time could be due to both the higher dose of drugs and the use of a front rather than a hind leg for IM injection as found in cats receiving preanesthetic treatment (personal author experience, unpublished data). Robert et al. reported an induction time that was statistically longer (2 min) in juveniles when compared to adults, and about 2.6 min longer during fall (when fat deposits are greater) than springtime. We did not appreciate any difference according to age and seasonality in induction time. Raccoons have a considerable amount of subcutaneous fat and its thickness varies between 3 and 5 cm. The amount of fat can interfere with anesthetics uptake, distribution and excretion, with the subcutaneous fat layer acting as a mechanical barrier preventing injections from reaching the skeletal muscle [32]. Animals in our study did not show any seasonal qualitative differences, mostly due to the fact that all the sterilizations occurred during late winter and early spring.

In our study, atipamezole did not influence recovery times as ‘end-palp’ and ‘end-ext’ were longer in KD group when compared to KM group. In contrast, Robert et al. reported that the time from atipamezole administration to when the animal had its head up was approximately 10 min; however, no surgery was performed in this study [23], and no additional anesthetics that may have contributed to a longer recovery (like observed in our study) were used [23]. Recovery times reported in our study were 25 min for the animals’ heads to be up and 49 min for the animals to stand up. The latter reported times are similar to the times published by Vogler, who reported 21 min for the animals to have their heads up without antagonization [31]. It was decided not to antagonize midazolam because the mean time from its administration and the end of anaesthesia was approximately 90 min, and we assumed that at this point the benzodiazepine would exert very little, or no, effect on the quality and duration of recovery as has been described in dogs [33].

As expected, during general anesthesia body temperature tended to decrease over time from an overall basal temperature of 36.3 to 35 ± 0.6 °C at 30 min. Cardiovascular parameters were maintained within physiological ranges in both groups; however, a greater tendency to respiratory acidosis was recorded in KM group. Despite normal PaO_2_ and SaO_2_, and a mild reduction in RR, respiratory depression (characterized by decreased pH and increased PaCO_2_) developed in all the animals enrolled in this study. Sufentanil CRI may have had an influence on our results, as it has been implicated in potentiating the depression of central chemoreflexes without affecting peripheral chemoreflexes. The depression of central chemoreflexes in sufentanil CRI occurs by reducing the tidal volume and RR in dogs (no data is currently available in raccoons) during anesthesia with isoflurane [34], in addition it will produce a dose-dependent depression of spontaneous ventilation in dogs under anesthesia with sevoflurane [35].

The authors are not aware of any study describing sevoflurane requirements in raccoons. The higher requirements detected in animals of KM group (1.6 ± 0.1 versus 1.4 ± 0.1% of KD group) may be a consequence of the lower degree of anesthetic depth achieved, as 42.9% of raccoons were scored as mildly anesthetized and required a half dose of the mixture as a top up. Nevertheless, it is likely that the concomitant sufentanil CRI might have contributed to decreased sevoflurane requirements in both groups, as reported in dogs during fentanyl infusion where a significant decrease of up to 41% has been described [36]. However, a simple extrapolation between different carnivorous species, especially among domestic or wild species, might lead to relevant errors in anesthetic dosing as reported in two different prosimian species [37]. Indeed, further studies with a larger sample size and different categories of animals would be needed to confirm our data and establish reference ranges for physiological parameters, helping with management of clinical cases in wild raccoons, as well as in anesthetic management during surgery.

## 5. Conclusions

The anesthetic protocols including ketamine coupled with either dexmedetomidine or midazolam can be recommended for chemical immobilization followed by general anesthesia of wild raccoons. Both protocols provided a reliable and fast immobilization and a safe induction of general anesthesia, which contributed to a stable surgical depth of anesthesia and to a smooth recovery. However, dexmedetomidine contributed to a greater synergistic effect than midazolam when combined with ketamine. Considerable attention should be given to the possible development of respiratory depression.

## Figures and Tables

**Figure 1 animals-10-02110-f001:**
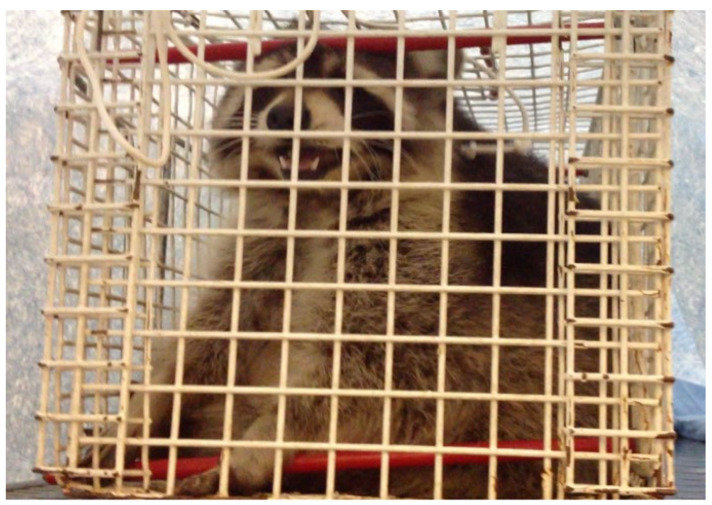
A wild raccoon placed into the squeeze cage to receive the intramuscular (IM) injection of the anesthetic mixture before gonadectomy.

**Figure 2 animals-10-02110-f002:**
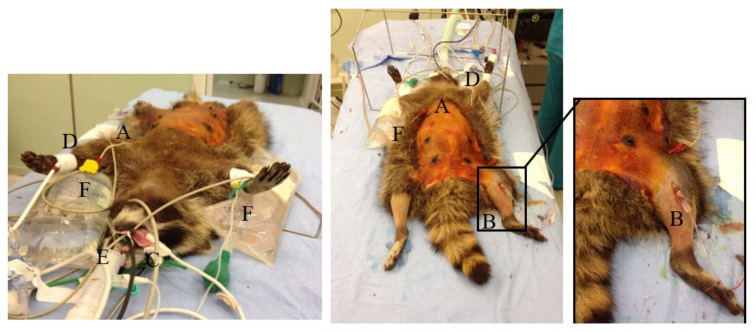
Wild female raccoon in dorsal recumbency, aseptically prepared for surgery and instrumented with monitoring devices: electrocardiogram (**A**), arterial line (**B**), pulse oximeter (**C**), cuff for oscillometric blood pressure measurement (**D**), oesophageal temperature probe (**E**), hot water bags (**F**). On the right side, a close up view of the arterial line placed in the femoral artery to measure blood pressure and to obtain arterial blood samples for blood-gas analysis during general anesthesia for gonadectomy can be seen.

**Figure 3 animals-10-02110-f003:**
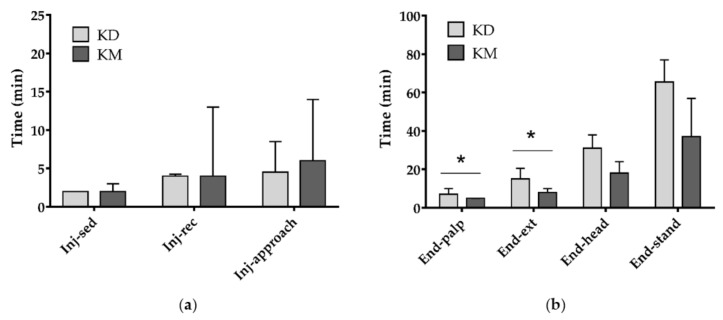
Description of some results related to times needed for induction and recovery observed in wild raccoons undergoing sterilization and randomized to receive one of two anesthetic mixtures IM: ketamine–dexmedetomidine (KD group, *n* = 10) or ketamine–midazolam (KM group, *n* = 7). (**a**) Times (minutes) from injection to the first sign of sedation (inj-sed), to recumbency (inj-rec) and to first manual approach (inj-approach) (**b**) Times (minutes) between end of anesthesia and appearance of palpebral reflex (end-palp), extubation (end-ext), lifting of the head (end-head) and standing (end-stand). Values are medians and interquartile ranges. KD = ketamine and dexmedetomidine; KM = ketamine and midazolam. Asterisks indicate significant differences between groups for each parameter (*p <* 0.05).

**Table 1 animals-10-02110-t001:** Descriptive Scoring System (DSS) used to categorize the quality and depth of anesthesia achieved in wild raccoons undergoing sterilization.

**Score**	**DEPTH OF ANESTHESIA**
1	Inadequate: Responsive to approach, no safe handling. Additional ½ dose is required
2	Mild: Purposeful response to stimulation (toe pinch). Additional ½ dose is required
3	Profound: Relaxed and unresponsive to stimulation (toe pinch)
**Score**	**MYORELAXATION**
1	Absent: Muscle tone in the pelvic and thoracic limbs, pedal reflex, persistent jaw tone, no relaxed tongue
2	Mild: Some muscle tone in thoracic limbs and pedal reflex, no relaxed tongue
3	Moderate: Some muscle tone in the pelvic or thoracic limbs, no jaw tone
4	Excellent: No muscle tone in pelvic and thoracic limbs, no jaw tone, relaxed tongue
**Score**	**CEPHALIC CATHETERIZATION**
1	Possible, evident limb retraction
2	Possible, mild limb retraction
3	Possible, without limb retraction
**Score**	**EASE OF INTUBATION**
1	Trachea intubation possible after drug administration (tongue retraction, swallowing reflex)
2	Trachea intubation possible without drug (no jaw tone, relaxed tongue)

**Table 2 animals-10-02110-t002:** Demographic data of wild raccoons undergoing sterilization and randomized to receive one of two anesthetic mixtures IM: ketamine–dexmedetomidine (KD group, *n* = 10) or ketamine–midazolam (KM group, *n* = 7).

Parameter	Group	
	KD	KM
Body weight (kg, mean ± SD)		
	7.00 ± 1.98	7.10 ± 0.81
Gender (*n*, %)		
Female Male	5 (50.0%)	4 (57.1%)
	5 (50.0%)	3 (42.9%)
Pregnant animals (*n*, % of females)		
No Yes	2 (40.0%)	2 (50.0%)
	3 (60.0%)	2 (50.0%)
Age * (*n*, %)		
YoungAdult	2 (20.0%)	1 (14.3%)
	8 (80.0%)	6 (85.7%)

* Age was defined based on physical characteristics such as external genitalia, body mass and teeth consumption [15].

**Table 3 animals-10-02110-t003:** Parameters related to the quality of anesthesia according to the DSS in wild raccoons undergoing sterilization and randomized to receive one of two anesthetic mixtures IM: ketamine–dexmedetomidine (KD group, *n* = 10) or ketamine–midazolam (KM group, *n* = 7). Values are numbers and percentages.

Parameter		Group		Total	*p* Value
		KD	KM		
Depth of anesthesia	Inadequate	0 (0.0%)	0 (0.0%)	0 (0.0%)	0.593
	Mild	2 (20.0%)	3 (42.9%)	5 (29.4%)	
	Profound	8 (80.0%)	4 (57.1%)	12 (70.6%)	
Muscle relaxation	Absent	2 ^a^ (20.0%)	1 ^a^ (14.3%)	3 (17.6%)	**0.003**
	Mild	0 ^a^ (0.0%)	3 ^b^ (42.9%)	3 (17.6%)	
	Moderate	1 ^a^ (10.0%)	3 ^a^ (42.9%)	4 (23.5%)	
	Excellent	7 ^a^ (70.0%)	0 ^b^ (0.0%)	7 (41.2%)	
Cephalic catheterization	Evident limb retraction	0 (0.0%)	1 (14.3%)	1 (5.9%)	0.338
	Mild limb retraction	1 (10.0%)	2 (28.6%)	3 (17.6%)	
	No limb retraction	9 (90.0%)	4 (57.1%)	13 (76.5%)	
Palpebral reflex	No	1 (10.0%)	1 (14.3%)	2 (11.8%)	1.000
	Yes	9 (90.0%)	6 (85.7%)	15 (88.2%)	
Response to tactile stimulus	No	8 (80.0%)	6 (85.7%)	14 (82.4%)	1.000
	Yes	2 (20.0%)	1 (14.3%)	3 (17.6%)	
Tongue relaxation	No	3 (30.0%)	5 (71.4%)	8 (47.1%)	0.153
	Yes	7 (70.0%)	2 (28.6%)	9 (52.9%)	
Ease of intubation	After propofol	3 ^a^ (30.0%)	7 ^b^ (100.0%)	10 (58.8%)	**0.010**
	Without other drugs	7 ^a^ (70.0%)	0 ^b^ (0.0%)	7 (41.2%)	

For each row, values followed by the same letter do not have statistically significant differences (at the level of 0.05). If no letters are present, the differences were not statistically significant. *p* values in bold denote statistical significance at the level of 0.05.

**Table 4 animals-10-02110-t004:** Effects of group, time and baseline values on physiological parameters monitored continuously in wild raccoons undergoing sterilization and randomized to receive one of two anesthetic mixtures IM: ketamine–dexmedetomidine (KD group, *n* = 10) or ketamine–midazolam (KM group, *n* = 7) and maintained under general anesthesia with sevoflurane and sufentanil constant rate infusion (CRI). Values are estimated marginal means ± standard error.

Parameter	Baseline Value *	Group		*p* Value			
		KD	KM	Group	Time	Group × Time	Baseline
Temp. (°C)	36.3	35.5 ± 0.1	35.6 ± 0.1	0.376	**0.007**	0.406	**<0.001**
SAP (mmHg)	115.4	108.2 ± 1.7	116.2 ± 1.7	**0.002**	0.901	0.906	**<0.001**
MAP (mmHg)	90.7	86.7 ± 1.5	92.4 ± 1.5	**0.011**	0.748	0.313	**<0.001**
DAP (mmHg)	71.3	68.2 ± 1.8	75.7 ± 1.8	**0.006**	0.621	0.332	**<0.001**
SpO_2_ (%)	97.8	98.6 ± 0.2	97.7 ± 0.2	**0.009**	0.703	0.286	**<0.001**
RR (breaths/min)	13.3	10.2 ± 0.6	10.4 ± 0.6	0.788	0.098	0.106	**<0.001**
HR (beats/min)	114.7	105.2 ± 2.5	109.3 ± 2.9	0.342	0.935	0.734	**<0.001**
etCO_2_ (mmHg)	40.6	33.1 ± 1.3	34.1 ± 1.4	0.888	0.340	0.419	**0.004**

* as it appears in the model. Bold values denote statistical significance at the level of 0.05. SAP = systolic blood pressure; MAP = mean arterial pressure; DAP = diastolic arterial pressure; SpO_2_ = arterial oxygen saturation of hemoglobin; RR = respiratory rate; HR = heart rate; etCO_2_ = end-tidal carbon dioxide partial pressure.

**Table 5 animals-10-02110-t005:** Blood-gas values in wild raccoons undergoing sterilization and randomized to receive one of two anesthetic mixtures IM: ketamine–dexmedetomidine (group KD, *n* = 10) or ketamine–midazolam (group KM, *n* = 7) and maintained under general anesthesia with sevoflurane and sufentanil CRI. Values are medians and interquartile ranges (IQR).

Parameter	Group				*p* Value
	KD		KM		
	Median	IQR	Median	IQR	
pH	7.26	(7.19–7.34)	7.11	(7.06–7.15)	**0.017**
PaCO_2_ (mmHg)	49.10	(40.50–56.50)	63.10	(60.40–70.10)	**0.033**
PaO_2_ (mmHg)	535	(473–562)	442	(423–562)	0.517
BE (mmol/L)	−4	(−5–−3)	−7	(−12–−7)	0.067
HCO_3_^−^ (mmol/L)	22.2	(20.3–24.6)	22.3	(17.7–22.9)	0.833
TCO_2_ (mmol/L)	24	(22–24)	24	(20–25)	1.000
SaO_2_ (%)	100	(100–100)	100	(100–100)	1.000
Na^+^ (mmol/L)	147	(145–149)	152	(150–154)	0.286
K^+^ (mmol/L)	3.8	(3.6–4.0)	3.4	(3.2–3.5)	0.143
iCa^2+^ (mmol/L)	1.26	(1.22–1.29)	1.13	(1.12–1.14)	0.286
Hct (%)	29	(22–38)	19	(16–21)	0.143
Hb (g/dL)	9.85	(7.50–12.90)	6.25	(5.40–7.10)	0.143

*p* values in bold denote statistical significance at the 0.05 level.

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
