# Peer review of "Gonadectomy in Raccoons: Anesthetic and Cardiorespiratory Effects of Two Ketamine-Based Pre-Anesthetic Protocols before Sevoflurane-Sufentanil"

_animals, 2020, doi:10.3390/ani10112110_

Round 1

Reviewer 1 Report

Dear Authors,

Your work is interesting but i would have some doubts about the numerical adequacy of the two groups:

How could you have ruled out the influence of variables related to sex and age with such a small sample of subjects enrolled in the study?

Did you perform a power analysis to determine the number of subjects to be enrolled in the study?

In this work I would have expected a greater increase in blood pressure in the KD group as occurs in all the studies that compare benzodiazepines with alpha 2 agonists, why did this not happen? Further, why was it not explained in the study?

One last consideration, why haven't you also antagonized midazolam? don't you think this may have affected the awakening times?

Author Response

Dear reviewer, please see attached file with responses to your questions.

Thank you.

Reviewer 2 Report

Dear authors,

Thank you for allowing me to review your work. I think it is a simple, clear and well-written article. It provides information on the efficacy and the cardiorespiratory effects of two anesthetic protocols for immobilization of raccoons. This is an interesting topic on which there is not much information published.

Just a few comments and suggestions about it:

Line 127. Please define IV

Figure 3. I suggest using the variable Time (min) instead of minutes.

I suggest rewriting the conclusions avoiding to include parts of the discussion or justification of the study:

Lines 333-341. I think this paragraph is more the justification of the study than the conclusion. I will delete it.

Lines 344-348. This part has been used at the beginning of the discussion.

Kind regards,

Author Response

(The authors gave the same response as above.)

Reviewer 3 Report

Please revise the following

I have detected some minor errors which are indicated below. Also same recommendation in order to improve your manuscript are proposed and detailed below.

Line 12. Please, use italic format for the scientific names i.e. Procyon lotor instead of Procyon lotor. Also check all the manuscript for this nomenclature rule.

Line 14. Add: .....influence on vital "clinical" parameters.

Line 15. Introduce the scientific names the first time it appears in the summary, later include only one of the option (scientific or common name) but not both. Also, use italic format if scientific names are used.

Line 44. Introduce the scientific names (and in italic format) the first time their appear in the introduction.

Line 47. Revise English: ... the raccoon is .... and safety and their? or it? sales ......

Line 47. ...sale and detention... (please clarify what do you mean with detention, do you refers to capture?).

Line 48. Add the link to the law in the references.

Line 49. Revise English: Raccoons are .... in the areas in which it? or they? was/were freed.

Line 50. This is a fact or an opinion?. If it is a fact, please add references.

Line 51. Ecological, economic and health ..... If any references in the literature, please add.

Line 55. Italic format should be used for Baylisascaris procyonis

Line 53 to 57. I recommend to add the words "Zoonoses" and "One Health" concept in order to increase the reference to this important veterinary input to the public health in the literature.

Line 58. Revise English:  in their respective jurisdiction territories? ...... authorized centers to? ...

Line 62. please clarify what do you mean with entity (do you refers to real number of animals in the population?). Not clear.

Line 72. Consider add: ... vital clinical? parameters, here and elsewhere in the text.

Line 83. Introduce the scientific names the first time it appears in the text, later include only one of the option (scientific or common name) but not both. Scientific names should be written in the nomenclature format.

Line 108. How many of the animals received this second administration? This condition was included in the results analysis?

Line 111. This is a self-reference [16]. Please provide also an external evidence of the DSS for depth anesthesia.

Line 131. The quantity of sevoflurane was not the same in all the animals? Please clarify this because the differences between groups is then affected by this variable in the depth of the anestesia during the maintenance period. Do you study this variable?

Table 3. 3rd line, 2nd column and last line. What is the meaning of a and b? Not clear in the foodnotes of the table or title.

Some comments:

For Analgesia, only a CRI of sufentanil was used. Due to it short activity, do you use any other analgesia for postsurgical time? Not mentioned in the text. Please, include if used.

Also, why any opioid was not used in the preanesthesia protocol?, do you think that its use may reduce the use of anesthetic drugs? Would this affect your results? Particularly the needs for second injection or the time for sedation and induction. Please clarify.

A higher description of the analgesia protocol for this procedures in wild raccoons should be included in the conclusion or, elsewhere in the text, in order to implement this protocols by other vets (readers).

Author Response

(The authors gave the same response as above.)

Round 2

Reviewer 1 Report

Dear authors, I am happy to accept your work with the corrections made,

Congratulations